# A systematic review of risk communication in clinical trials: How does it influence decisions to participate and what are the best methods to improve understanding in a trial context?

**Maeve Coyle, Katie Gillies** *

Health Services Research Unit, University of Aberdeen, Aberdeen, United Kingdom

* k.gillies@abdn.ac.uk

## Abstract

### Background

Effective risk communication is challenging. Ensuring potential trial participants' understand 'risk' information presented to them is a key aspect of the informed consent process within clinical trials, yet minimal research has looked specifically at how to communicate probabilities to support decisions about trial participation. This study reports a systematic review of the literature focusing on presentation of probabilistic information or understanding of risk by potential trial participants.

### Methods

A search strategy for risk communication in clinical trials was designed and informed by systematic reviews of risk communication in treatment and screening contexts and supplemented with trial participation terms. Extracted data included study characteristics and the main interventions/findings of each study. Explanatory studies that investigated the methods for presenting probabilistic information within participant information leaflets for a clinical trial were included, as were interventions that focused on optimising understanding of probabilistic information within the context of a clinical trial.

### Results

The search strategy identified a total of 4931 studies. Nineteen papers were selected for full text screening, and seven studies included. All reported results from risk communication studies that aimed to support potential trial participants' decision making set within hypothetical trials. Five of these were randomised comparisons of risk communication interventions, and two were prospectively designed, non-randomised studies. Study interventions focused on probability presentation, risk framing and risk interpretation with a wide variety of interventions being evaluated and considerable heterogeneity in terms of outcomes assessed. Studies show conflicting findings when it comes to how best to present information, although numerical, particularly frequency formats and some visual aids appear to have promise.

**Data Availability Statement:** The data underlying the results presented in the study are available from the published papers. Included studies available here: Reference 22 - DOI: 10.1177/

0092861506040003O2 Reference 23 - DOI: 10.1542/peds.2009-1796 Reference 24 - DOI: 10.1177/1740774515585120 Reference 25 - DOI: 10.1017/S1357530902000558 Reference 26 - DOI: 10.1097/00000539-200302000-00037 Reference 27 - DOI: 10.1186/1472-6947-10-55 Reference 28 - DOI: 10.1177/014107689008300710.

**Funding:** This work was supported by personal fellowship award (to KG) from the Medical Research Council's Strategic Skills Methodology Programme. The Health Services Research Unit is supported by a core grant from the Chief Scientist Office of the Scottish Government Health and Social Care Directorates. The views and opinions expressed therein are those of the authors and do not necessarily reflect those of the Chief Scientist Office, MRC or the Department of Health.

**Competing interests:** The authors have declared that no competing interests exist.

## Conclusions

The evidence base surrounding risk communication in clinical trials indicates that there is as yet no clear optimal method for improving participant understanding, or clear consensus on how it affects their willingness to participate. Further research into risk communication within trials is needed to help illuminate the mechanisms underlying risk perception and understanding and provide appropriate ways to present and communicate risk in a trial context so as to further promote informed choices about participation. A key focus for future research should be to investigate the potential for learning in the evidence on risk communication from treatment and screening decisions when applied to decisions about trial participation.

## Introduction

Clinical trials are now widely accepted as the gold standard of evidence-based medicine for determining treatment effects [1]. The importance of recruiting adequately informed individuals to participate in clinical trials is paramount. However many studies have demonstrated that participants approached to take part and those consented to participate in trials have a limited understanding of key aspects of the trial [2]. One of the key areas to consider when presenting information to potential participants is the information on potential risks and benefits. The ethical principles for medical research involving human subjects, enshrined within the Declaration of Helsinki, state that 'each potential subject must be adequately informed of the anticipated benefits and potential risks of the study' [3]. This is echoed by guidelines for good clinical practice, which state that all the information provided to participants should include explanations of the 'reasonably foreseeable risks or inconveniences', expected benefits, and where there are no clinical benefits to the participants, they must be made aware of this' [4]. However, mechanisms to operationalise the provision of such information are not provided in the guidance.

Risk communication can be defined as communication with individuals that addresses knowledge, perceptions, attitudes and behaviour related to risk, and risk itself can be defined as the probability that a hazard will give rise to harm [5, 6]. A correct understanding of risk therefore depends upon an accurate understanding of probabilities, a feat that is determined by several influencing factors, such as individual numeracy levels and cognitive abilities, but not least by the methods used to present probabalistic information [7]. There is a substantial amount of literature that focuses on risk communication with regard to public health messages, health behaviour, and treatment and screening decisions for patients [8–11]. Speigelhalter et al have shown that probabilities are 'notoriously difficult to communicate effectively' to lay audiences in various contexts, including health [12]. Yet minimal research has looked specifically at how to communicate probabilities within information provided to support decisions about trial participation (or not). In a trial context uncertainties relating to interventions will usually be greater purely by the nature of the trial endeavour—to generate evidence about benefit and harm.

Understanding, or more often mis-understanding, of risk information related to trials has been shown to influence decisions about participation in a range of trials, with those prepared to accept risk more likely to participate [13, 14]. Decisions about trial participation are inherently different from decisions about treatment. For example, one of the main influences on clinical trial participation is conditional altruism [13]. Conditional altruism is the concept that

participation in the trial will benefit society but there must be a benefit (which is influenced by perception of risk) for self. Conditional altruism does not exist for decisions about treatment and as such it is important to understand how potential trial participants understand risk in a trial participation context. Additionally, trade-offs between risk and benefit in a trial involve layers of complexity in addition to those for treatment such as: loss of control over which treatment they receive; and potentially greater uncertainties, as often participants have to consider the risks and benefits of a minimum of two competing treatments. Existing studies in the domain of informed consent for clinical trials have repeatedly highlighted significant discrepancies between actual risk and participant interpretation of risk to themselves, or their child, in taking part in a trial [15, 16]. Participants frequently underestimate risks, leading them to believe that there would be little to no risk involved in trial participation. This pronounced lack of understanding strongly suggests the need for better communication about trial aims and design, particularly when it comes to the inherent risks, however small, that are almost always present in taking part in a clinical trial [15]. The intrinsic nature of trials means there is much unknown information and communicating probabilistic information in this context is more challenging as the layers of risk are greater, for example the risk of undertaking a trial as opposed to treatment, the outcome risks, and the risk of randomisation to a drug, procedure or placebo [17].

Preliminary findings from our group have shown that stakeholders have varied preferences about how probabilistic information relevant to trial participation (e.g. estimates of the likelihood of benefit and/or harm associated with trial interventions) is communicated [18]. In addition, a pilot study exploring decision support for trial participation decisions highlighted that patients' preferences for risk information differed in a trial context compared to a treatment context [19]. Existing research on methods to present probabilistic information to improve patient understanding and decision making about treatment and screening decisions could provide valuable insights for enabling effective risk communication in the context of informed consent for trials [20]. Yet, surprisingly, the methods shown to be effective to improve understanding of probabilistic information are not routinely employed in participant information leaflets for trial participation [17].

A small number of studies have evaluated methods for presenting 'risk' in patient information leaflets for clinical trials. However, these studies have not been analysed together to allow judgements about optimal methods of presentation. This warrants further investigation both at the level of understanding and on the decision to participate (or not) in the trial. To address this, this study aimed to systematically review the literature focusing on presentation of probabilistic information within the informed consent process for trials. We focused our search on comparative effectiveness studies that tested interventions which varied the presentation of probabilistic information and the effects on potential trial participants' understanding and/or the decision to participate.

## Methods

### Inclusion criteria

Evaluative studies using qualitative methods that investigated the methods for presenting probabilistic information to potential trial participants during the informed consent process for a trial were considered eligible. Specific study designs could include randomised controlled trials, case series, and prospective cohorts. Interventions that focused on optimising understanding (or another plausible outcome linked to decision making for trial participation) of probabilistic information within the context of a clinical trial were included. We chose to include studies of both real and hypothetical decisions about trial participation.

## Exclusion criteria

Papers or articles that present findings on risk communication in a treatment or screening context or consider the decision to participate in research studies that are not RCTs were excluded. Studies investigating participants' perceptions of receiving risk communication as part of the RCT decision process (which may include studies using methods such as interviews, focus groups and other methods) were not included.

## Search methods for identification of studies

A search strategy for risk communication in clinical trials was designed in collaboration with a Senior Information Specialist (skilled in developing and running search strategies to identify relevant scientific literature) and informed by systematic reviews of risk communication in treatment and screening contexts and supplemented with trial participation terms. The search strategy is available on request. Four data bases were searched. Embase was searched from 1980 to 2019. Ovid MEDLINE(R) Epub Ahead of print, In-Process & other Non-Indexed Citations, Ovid MEDLINE(R) Daily and Ovid MEDLINE(R) was searched from 1946 to May 10[th] 2019. PsycINFO was searched from 1987 to May week 2 2019. Finally, CINAHL was searched from 1998 to 2019. No restrictions on language were imposed.

## Screening and selection of studies

One author (MC) screened all articles identified within the database searches. Duplicate screening was carried out by one other author (KG) on a random sample (10%) of the search output. Papers were assessed at title and abstract level according to the eligibility criteria, and differences of opinion were resolved by discussion between MC and KG. Nineteen full text papers were identified for further investigation, and of these seven studies were deemed eligible for inclusion and progressed to data extraction procedures.

## Data collection and analysis

The seven studies were summarised by study characteristics (see details below) and presented in tabular form. Due to the heterogeneous nature of the interventions and/or outcomes reported a meta-analysis was not appropriate. This review is therefore presented in a descriptive narrative form with studies grouped first by design of the embedded study (RCT, non-randomised) and then by content of intervention i.e., probability presentation, risk framing, risk intervention. This structured framework to present narrative findings has been recently proposed by Rowlands et al 2018 [21].

## Data extraction and management

Data were extracted independently by two reviewers (MC & KG). The following summary features of the host trial (i.e. the trial the potential participants were being asked to consider participation in) for each study were summarised in table form: study design; study aim; author details; year and journal of publication; population demographics; sample size; phase of trial; intervention(s). Specific details on the intervention(s) being evaluated (i.e., risk communication tools), embedded study results and associated outcomes were extracted. These included: comparative methods of disseminating probabilistic information to potential trial participants using different communication tools/aids; mode of intervention delivery (i.e., paper, computer, verbal); study outcomes to be extracted; cognitive outcomes (i.e., potential trial participant comprehension of probabilistic information and subsequent risk perception); affective outcomes (i.e., participant preferences and/or satisfaction with communication method, and

level of decisional conflict and concern); and behavioural outcomes linked to trial participation (i.e., willingness to participate in clinical trial).

# Results

## Study selection and summary characteristics

The search strategy identified a total of 4931 studies. Full text papers for 19 potentially eligible studies were sourced, and following full text screening a further 12 studies were excluded from the review (Fig 1). The included seven studies all reported results from studies set within hypothetical randomised controlled trials [22–28]. To provide an example of how this embedded evaluation is operationalised, the studies asked participants to imagine they were being recruited into a clinical trial, provided brief information about the hypothetical trial (such as clinical population, intervention, comparator, outcomes, etc), then provided various formats of risk communication (such as verbal or numerical descriptors) followed by assessment of relevant outcomes.

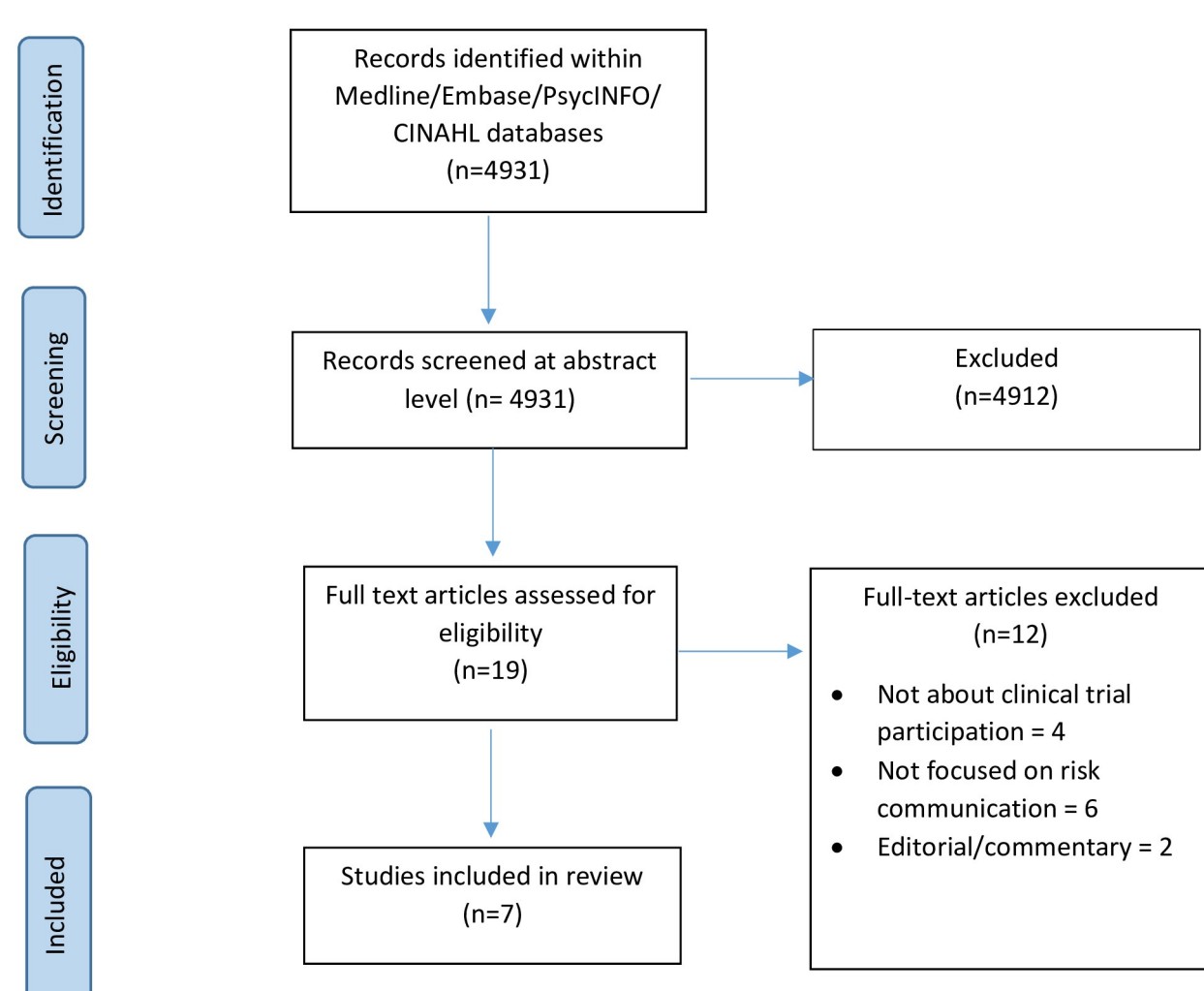

**Fig 1. The PRIMSA diagram details our search and selection process applied during the literature review.**

Table 1. Summary characteristics of hypothetical host trials.

| Trial characteristic | Hypothetical RCTs (n = 7) |
|---|---|
| **Clinical setting** | |
| Dermatology | 1 |
| Neurology (pain x 2, ALS) | 3 |
| Cardiology | 1 |
| Oncology | 1 |
| Surgical site infection | 1 |
| **Types of interventions** | |
| Drug | 6 |
| Non-drug | 1 |
| **Trial design** | |
| Cohort | 1 (Sutherland) |
| Two groups | 2 (Berry, Kim) |
| More than two groups | 4 (Cheung, Schwartz, Tait, Treschan) |
| **No of centres** | |
| Single centre | 5 |
| Multicentre | 0 |
| Unknown | 2 |
| **No of participants** | |
| <500 | 5 |
| 500–1500 | 1 |
| >1500 | 1 |
| Median no of participants (range) | 240 (50–4885) |

The seven included studies had various designs: five were randomised comparisons of risk communication interventions considering participation in hypothetical RCTs; and two were prospective, non-randomised studies, one being a comparative cohort study (three groups) and the other a single cohort. The included studies spanned a range of clinical settings. Three of the included studies were trials in neurological settings, and the other four were within dermatology, cardiology, oncology and surgery. Only one study was set within a trial considering a non-drug intervention, where the other six were identified as trials testing drug-based interventions. All of the included studies had at least two arms as part of their hypothetical trial design. Six of the seven studies reported trials where an individual was considering consenting for themselves and one study included only parents who were considering participation for their child. Most studies were single centre, however two of the studies did not specify the number of centres involved. The number of participants in the embedded studies ranged from 50 to 4885 with a median of 240. (Table 1).

The final seven studies were grouped according to the study design (i.e., RCT or prospective cohort) and the topic of the described intervention: 'probability presentation' (22, 23), 'risk framing' (24, 25, 26, 27), and 'risk interpretation' (28). The studies are presented alphabetically based on these similar characteristics under their category headings (Table 2). Further information on each study detailing intervention content, mode, and outcome are presented in Table 3.

## RCTs of interventions to explore risk communication in RCTs

**Probability presentation interventions.** One study was identified that used a randomised design to investigate different probability presentations in the context of risk communication

**Table 2. Catalogue of included studies by study design and real or hypothetical RCT setting.**

| Category | Risk communication study design | Real RCTs | Hypothetical RCTs |
|---|---|---|---|
| A | RCTs of interventions to explore risk communication in RCTs | N/A | Berry<br>Kim<br>Schwartz<br>Tait<br>Treschan |
| B | Prospectively designed, non-randomised studies of interventions to explore risk communication within RCTs | N/A | Cheung<br>Sutherland |

in clinical trials. Berry & Hochhauser (2006), compared European Union (EU) verbal descriptors only versus verbal descriptors and their associated numerical values (e.g., Common (EU equivalent = 1–10%)) [22]. Participants were asked to imagine they had been approached to take part in a clinical trial and given a booklet detailing the possible side effects of a new drug (versus nothing) for a skin condition and were asked to complete a questionnaire (N = 96, 48 in each arm). When asked to rate on a scale from 1 to 6 (p = 0.03), those who received only verbal descriptors were significantly less satisfied with the information than those who also had the numerical values. Participants in the verbal descriptors only group also perceived the risk to health to be higher (p<0.0001) and the benefit to be lower (p = 0.03), and were significantly less likely to participate in the trial (p = 0.01). When asked to make probability estimates for experiencing side effects, the verbal only group estimated these approximately three times higher than the combined group. When asked to consider the main reason for participating in the trial, participants in both groups reported long term relief/possible cure and the main reason for not participating was fear of side effects. There were no significant differences between the reasons listed by the two groups.

**Risk framing interventions.** Four of the included studies employed randomised designs to explore risk framing in communication within clinical trials. Kim *et al* (2015) recruited 584 participants to investigate the language framing of benefit statements within a hypothetical trial for amyotrophic lateral sclerosis. An online survey administered one of two statements within a consent form to participants; either 'there is some but very small chance that you might benefit' (control group n = 290), or 'it is not guaranteed you will benefit' (intervention group n = 294) [24]. The intervention group had a slightly greater, but not significant, willingness to participate in the trial as scored on a 10 point scale (p = 0.11). However, the average estimate of the likelihood of their condition improving was significantly higher in the intervention group than in the control group (p<0.0001).

Schwartz & Hasnain (2002) explored the effects of gain and loss framing on risk perception and attitude by randomising 284 participants to one of three groups receiving a consent form about a trial for a new cholesterol lowering drug [25]. One group were given information where benefits were framed in terms of gains (e.g., 'Out of 100 people whose lives would likely be cut short by heart disease and begin taking this drug, we expect that 95 will show substantial improvements in their chance of survival and 5 will show no improvement in survival', n = 98), the 'loss' group received benefit information framed as losses (e.g.,. 'Out of 100 people whose lives would likely be cut short by heart disease and begin taking this drug, we expect that 5 people will go on to die from heart disease, and 95 people will reduce their chance of death', n = 93), and the third group were given information where both framings were presented (e.g.,. 'Out of 100 people whose lives would likely be cut short by heart disease and begin taking this drug, we expect that 5 people will show no improvement and will go on to die from heart disease, and 95 people will substantially improve their chance of survival and reduce their chance of death', n- = 93). The majority of participants (59%) chose to take part in

**Table 3. Types of intervention(s) reported in included studies.**

| Author/Date | Content | Mode | Outcome |
|---|---|---|---|
| *RCTs of interventions to explore risk communication in RCTs* | | | |
| *Probability presentation interventions* | | | |
| Berry & Hochhauser 2006 | Intervention communicates risk for the 'experimental' drug only<br>Two experimental conditions: probability of side effects described using verbal descriptors, or verbal descriptors with associated numerical ranges. i.e.<br>• Common (EU equivalent = 1–10%)<br>• Uncommon (EU equivalent 0.1–1%)<br>• Rare (EU equivalent = 0.01%-0.1%) | Written four-page questionnaire booklet | • Satisfaction with the information; perceived risk to health from taking the drug; perceived effectiveness of the drug; how beneficial for their health it would be if they took part in the trial; and how likely it was that they would participate. (assessed using a 6-point unipolar Likert rating scale)<br>• Estimation of the probability of their experiencing each side effect as a percentage, between 0% and 100%.<br>• Main reasons for taking part or not |
| *Risk framing interventions* | | | |
| Kim *et al* 2015 | Intervention communicates risk for the 'experimental' drug only<br>One of two statements: in 'no guarantee' group, the likelihood was described as 'It is not guaranteed you will benefit'; in 'control' group likelihood described as 'There is some but very small chance that you might benefit' | Online survey | • Willingness to participate in trial on a 10-point scale, from 'would not consider at all' = 0, to 'definitely would consider' = 10.<br>• Likelihood that ALS would improve from being in this study, from 0% to 100%. |
| Schwartz & Hasnain 2002 | Intervention communicates risk for the 'experimental' drug only<br>Information in each group's consent form was identical except for the second paragraph describing the probable risks and benefits of the new drug.<br>In the 'gain' group, benefits were framed in terms of gains, benefits were framed as losses in the 'loss' group, and in the 'both' group, both framings were presented i.e.:<br>'Anyone taking this drug has a small risk of a severe allergy that could result in death. Out of 100 people whose lives would likely be cut short by heart disease and begin taking the drug, we expect that 5 people will show no improvement and will go on to die from heart disease (loss), and 95 people will substantially improve their chance of survival and reduce their chance of death (gain)' | Paper based | • Riskiness of participation in the clinical and riskiness of non-participation in the clinical trial on a category rating scale from 1 (not at all risky) to 10 (extremely risky)<br>• Willingness to participate in trial (yes or no) |
| Tait *et al* 2010 | Intervention communicates risk for both interventions (drug)<br>Three different risk/benefit message formats (text, tables or pictographs), and the presence or absence of a risk severity graphic.<br>Risks (itching and slowed breathing) and benefit (pain relief) were communicated in one of the three different formats.<br>Comparing risks and benefits between drugs A and B: absolute risk of occurrence for drug A was presented; information for drug B was presented as incremental risk increase or decrease. | Online survey | • Verbatim understanding (the ability to correctly report the actual risk and benefit frequencies of drugs A and B)<br>• Gist understanding (the ability to identify the essential meaning about the observed differences between the risks and benefits of drugs A and B)<br>• Perceptions of the risks and benefits of drugs A and B e.g. 'how worried would you be about your child experiencing pain after surgery?'. Also perceptions of frequency and severity of side effects, scored using 1–11 interval scales from e.g. 'not at all likely/worried' to 'extremely likely/worried' etc.<br>• Perceptions of the risk/benefit communication format |
| Treschan *et al* 2003 | Intervention communicates risk for both the control and the treatment group<br>Three versions of study protocol: 'control' involved little if any risk or pain; 'pain' required additional procedures that were described as provoking considerable pain and discomfort; and 'risk' involved additional procedures that were described as inducing risk of injury | Study protocol and informed consent document | • Willingness to participate (yes or no)<br>• Understanding of risks involved in participation i.e. asked to mark the statement they found most applicable: A. 'Participation in this study is not associated with additional risks, discomfort or pain', B. 'Participation in this study is associated with additional risks, but does not cause any discomfort or pain', and C. 'Participation in this study is not associated with additional risks, but might cause discomfort or pain'.<br>• Factors that influenced willingness to consent |
| *Prospectively designed, non-randomised studies of interventions to explore risk communication within RCTs* | | | |
| *Probability presentation interventions* | | | |

*(Continued)*

**Table 3.** (Continued)

| Author/Date | Content | Mode | Outcome |
|---|---|---|---|
| Cheung *et al* 2010 | Intervention communicates risk for the new medication only<br>Three formats of risk presentation: frequency, percentage and verbal descriptors. The verbal description followed the EU guideline on drug labelling; risk levels of ≤0.01%, >0.01% to 0.1%, >0.1% to 1%, >1% to 10%, and >10% were described as 'very rare', 'rare', 'uncommon', 'common' and 'very common' respectively.<br>Card 1 showing information about side effects of a new medication for pain relief in one of 6 ways of risk presentation.<br>Card 2 with the same risk information presented in all three formats (in the same sequence in severity) | Paper based | • Willingness to participate in trial after card 1 presentation, and then willingness to participate in trial after card 2 presentation;<br>• A change in decision would indicate a potential problem in the initial format.<br>• Preference for risk communication<br>• Understanding of EU descriptors: which of the five (from 'very rare' to 'very common') best describe the frequency of 1 out of 40, 1 out of 4,000, 1 out of 5, 1 out of 200 and 1 out of 20,000. |
| *Risk interpretation interventions* | | | |
| Sutherland *et al* 1990 | Intervention communicates risk for the 'experimental' drug only<br>Patients asked to underline statements in the consent form that were pertinent to making a decision about participating in the study.<br>Three statements about the likelihood of certain events occurring were given; 'itchy, red skin rashes are unlikely to occur', 'a particular type of cancer responds to radiation treatment in 10% of cases', and 'nausea and vomiting occurs in 45% of patients' | Paper based | • Willingness to participate in trial<br>• Understanding of three statements describing probability of an event occurring.<br>• Preferences for the way potential benefits and risks or side effects of therapy are described<br>• Preference for verbal and/or numerical descriptors of probability |

the trial when outcomes were framed as losses, while only 35% of the 'gain' group chose to participate. When both framings were presented, 62% of participants chose to participate, making a similar choice to the 'loss' group. When it came to perceiving riskiness of participation, the 'gain' group were more likely to rate this as riskier than non-participation (66%) compared to the 'loss' group (55%). For the 'both' group, the results were again similar to the loss condition, with 52% reporting trial participation as riskier than not. Respondents in the gain condition rated participation as significantly riskier (on a 10 point scale) than those in the loss condition (p<0.05), and respondents in the loss condition rated non-participation as significantly riskier than those in the gain condition (p<0.05). There was a significant association between domain (gains vs loss) and relative riskiness of participation vs non-participation (p<0.05).

In the study by Tait *et al* [26] 4685 parents were asked to consider their child was being randomised into a trial testing two drugs for post-operative pain, one a standard treatment and the other proven in adults but not in children. The risks and benefits of the two drugs were presented in absolute terms with comparisons presented as incremental changes. Four scenarios that provided different risk/benefit trade-offs were developed and considered: one benefit and 2 risks (a minor and a major), which were varied for Drug B across each scenario but remained static for Drug A. There was one scenario with no trade off, where there was an increase in benefit as well as risk reduction (n = 1171), whereas the other three included a loss of benefit but gains in risk reduction (n = 1184, n = 1196, n = 1134). Overall the study showed that parents who received the 'no trade off' (i.e., improvements across benefit and risk) scenario had both improved gist (defined as 'ability to identify the essential meaning about the observed differences' and measured using 4 items where ≥3 correct answers were required, p<0.01) and verbatim understanding (defined as understanding or knowledge to 'correctly report the actual risk and benefit frequencies' and measured using 7 items where ≥5 correct answers were required, p<0.01). The no trade off scenario also enabled parents to correctly perceive the potential benefits as greater, risks as lower, (p<0.01) and to be more likely to agree to their child participating in the trial (measured using an 11 point scale) compared to

the other three groups (p<0.01). Taken together these results suggested the no trade off scenario offering multiple gains resulted in a higher level of scrutiny compared to when only reductions in risk were presented.

Treschan *et al* (2003) randomised 148 participants to one of three versions of a study protocol to examine how understanding of risk and discomfort associated with a clinical trial influences patients' decision to participate [27]. The proposed trial was comparing peri-operative oxygen (30% vs 80%) to reduce the risk of surgical site infections. The control group received a version of the protocol that stated there would be little if any risk or pain involved in participating (n = 47), the 'pain' group were told that there would be additional procedures that would cause considerable pain and discomfort (e.g., dressing of wounds, cannulation, blood samples, n = 51)), and for the 'risk' group procedures were described as having a high risk of injury (e.g.,. extra oxygen is dangerous, risks of cannulation, risk of blood samples, etc, n = 50). Participants in the control group were more willing to participate in the trial (64%), with significantly fewer consenting in the risky (26%) and painful (35%) groups (p<0.001). There were no significant differences in understanding of the level of risk or pain for the three groups (p = 0.884). Those who correctly understood the risk or pain described in the protocols were twice as likely to consent to participation in the trial (49% vs 24%, p = 0.003).

## Prospectively designed, non-randomised studies of interventions to explore risk communication within RCTs

**Probability presentation interventions.** Of the two non-randomised papers that met the inclusion criteria, Cheung et al (2010) is the only study that investigated probability presentation within risk communication for clinical trials [23]. This study implemented a cognitive experiment (N = 240) and preference survey about risk within a hypothetical trial for pain medication for arthritis. The intervention used a factorial design to study the impact of three formats (frequency (n = 82), percentage (n = 80) and verbal descriptors (n = 78)) and two sequences on willingness to participate and likelihood to change one's willingness after given additional information. Participants were presented with information in one of the six combinations. Participants were given a card that showed information about side effects of a new medication for pain relief in one of six ways of risk presentation, and then were asked whether they would be willing to take part in the trial. They were then presented with a second card, with the same risk information presented in all three formats being studied. A change in decision would indicate a potential problem in the initial format given to participants. There was no difference in willingness to participate in the trial across all presentations (p = 0.886), and there was also no difference in the likelihood of a participant changing their mind after being given the information in additional formats (p = 0.529). After reading card 2, the proportion of participants in each group showing a willingness to participate increased significantly (p<0.05). With regardto presentation preferences, 43% of participants preferred the frequency format, 32% preferred percentages, and 25% preferred the verbal descriptors.

**Risk interpretation interventions.** The remaining non-randomised study (Sutherland et al, 1990) explored risk interpretation within a consent form for a hypothetical drug trial for cancer [28]. All participants (N = 50) were given a consent form and asked to underline statements that were important to them in terms of making a decision about participating in the trial. They were also asked to indicate if their chosen statements were positive or negative. A questionnaire including preferences for probability descriptors (verbal or numerical) was also administered. Of those who refused to take part in the hypothetical trial, 70% noted only the potential for risk, 10% only for benefit, and the remaining 20% noted both risk and benefit information as important for their decision. Just 33% of those who 'consented' identified only

risks, 27% noted only benefits, and 30% noted both risk and benefit. The remaining 10% identified neither as important to their decision. One third of participants were unable to identify the correct interpretation of the 'unlikely' verbal descriptor, and 54% gave an incorrect interpretation of '10% response rate' meaning. When it came to preferences for benefit descriptors, 16% of patients preferred words, 34% numbers, 48% both and 2% other. For risk communication preference the results were very similar; 16% verbal, 28% numerical, 48% both and 2% other.

## Discussion

The study is one of the first to systematically review the published evidence on methods for communicating risk to potential trial participants during the informed consent process. It has examined and summarised the existing evidence about how risk information is perceived by potential participants and highlights how these factors may influence decisions to participate in a clinical trial context. Only seven studies were identified that have investigated aspects of communicating risk information in a clinical trial setting. Whilst the majority of studies were randomised comparisons, we also identified 2 non-randomised evaluations. Given the heterogeneity of the interventions investigated in the included studies and the variability in outcomes reported, a meta-analysis of these studies was not possible. This work therefore highlights the need for the rigorous development and evaluation of interventions to improve the presentation and communication of risk information for potential trial participants.

One of the studies investigated probabilistic presentation methods and demonstrated that numerical formats appear to be better at communicating risk to potential trial participants, when compared to text [22]. Participants receiving verbal descriptors alone were less likely to consent to take part in a trial and were less satisfied with the information, perceiving risks of side effects to be much higher than participants receiving both numerical and verbal descriptors. Similar findings can be seen in a review on communicating with patients about evidence (for treatment decisions), which illustrated that patients have a better understanding of risk if probabilistic information is presented numerically rather than verbally [29]. It is worth considering that studies in a treatment setting have shown that using visual aids such as pictographs or bar charts to present event rates may aid accurate understanding of probabilities, and they can help reduce several biases including framing effects [30]. There are many variants of visual aids however, and how these are utilised and understood by potential trial participants warrants more investigation using the best practice examples from treatment decision making as a starter.

The second study (Cheung et al, 2010) looking at probabilistic presentation found no difference in willingness to participate between frequency, percentage and verbal conditions; however, it did find a strong preference for numerical presentations over verbal descriptors, particularly for frequency formats [23]. Research by Price et al (2007) found that frequency statements are generally better understood by participants compared to ratios or percentages [31]. An important finding from this study highlighted major errors in correctly matching EU descriptors of risk to associated frequencies, findings echoed by the other study which looked at risk interpretation showing a large proportion of participants were unable to correctly interpret verbal descriptors or percentage formats [28]. A number of studies have demonstrated that many lay persons are unable to understand basic aspects of probabilities that are essential to risk understanding, nor to comprehend the concept of risk in general [32, 33]. This poses a challenge to effective risk communication and demonstrates a need for improved methods for better informed consent within the context of clinical trials.

The Sutherland *et al* (1990) study found that the majority of non-consenters to the trial noted only the potential for risk in the provided information, whereas the information was interpreted very differently by consenters where a minority saw only risks, and many perceived benefits instead [28]. A qualitative study into patient decisions about taking part in an epilepsy treatment trial noted that participant decision making was most commonly influenced by their perception of harm and benefit [34]. Those who agreed to take part usually saw the risks involved as acceptable, in this case because of the 'tried and tested' nature of treatments. However, the non-consenters viewed participation as 'an unknown quantity' and defined the risks of being randomised to an unsuitable drug as being too high or not in their best interest [34].

When it came to studies looking at risk framing, the results were mixed. The study by Kim *et al* found no significant difference in willingness to participate in the trial, although participants in the intervention group (no guarantee for benefit statement) were much more likely to believe that their condition would improve [24]. When benefits were framed as losses participants were more likely to take part in the trial, and when benefits were presented as both losses and gains, participants seemed to respond similarly to the loss group, suggesting that loss framing had more impact on decision making than gain, where perceived risk was higher [25]. However, many of the statements used in this study were vague and uninformative, putting into question what understanding participants had in relation to these statements in addition to willingness to participate. Conversely, Treschan *et al* found that when outcomes were framed as gains the majority of participants were less likely to participate [27]. Earlier research by Tversky & Kahneman (1981) on framing and the psychology of choice demonstrated that framing outcomes in terms of gains does indeed generate risk-averse choices, which could translate to, for example, a decreased willingness to participate in a clinical trial [35]. A more recent study highlighted the introduction of potential bias in decision making about trial participation when the effects of language framing are not addressed [36]. This study explored whether presenting health care decisions as 'opportunity' rather than 'choice' biased individuals' preferences in the context of trial participation for cancer treatment. They found that a 'choice' frame, where all treatment options are explicit, is less likely to bias preferences [36]. It is therefore of paramount importance that information given to participants include neutral statements, or at a minimum balanced statement about participation or not, so as not to unduly manipulate or 'nudge' decisions in ways that are not consistent with the individual's values and preferences [18].

Five out of the seven studies included in this review only communicated risk information about the 'experimental' treatment [22–25, 28]. Two studies communicated risk information about both the intervention and its comparator or indeed both active interventions [26, 27]. Given that decision making about clinical trials is complex and requires trade-offs between both (or all) options and therefore presenting risk (and benefit) information on these options would be important to support fully informed choices. This should be acknowledged and explored in future studies.

Complex language and details included in participant information leaflets (PILs) and consent forms for trials can be difficult for some people to comprehend properly and may engender more confusion than understanding of trial processes, including risks [37]. An analysis of PILs used in clinical trials by Gillies *et al* (2011) found that: explaining trial processes; presenting probabilities; and expressing values, were consistently poor across all PILs when assessed using an informed consent evaluation instrument [17]. These information leaflets clearly need to be improved to encourage higher quality decision making when it comes to trial participation. It is also clear that potential trial participants continue to have significant deficits in their recall and understanding of trial related information, and that such information is often not presented in a comprehensive way that optimises participant understanding [38, 39]. The

recent study by Gillies et al (2014) explored whether patient information leaflets (PILs) were able to effectively support decision making about trial participation [17]. They found that information that demonstrated support for good quality decision making in other contexts was lacking in PILs for UK clinical trials. In particular, the section on 'presenting probabilities' was almost always absent, despite its proven importance for supporting good quality decision making [17].

Whilst not a focus of this review it is important to point out that none of the included studies reported including patients or the public as partners in the research to identify what the content and/or presentation of the information should be for the studies. Also, no input was sought with regard to whether the outcomes being evaluated were appropriate and meaningful for patients faced with decisions about trial participation.

Lessons from effective risk communication in a treatment and/or screening context can provide examples of best practice that could be used for those developing PILs for patients considering clinical trial participation. A systematic review on risk communication published since has shown that visual aids, such as icon arrays and bar graphs, improved both understanding and satisfaction [40]. Interestingly, this review showed that presenting absolute risk reduction was better at maximising accuracy and less likely to influence decisions. The presentation of information on numbers needed to treat reduced understanding. This review also concluded that due to the quality and heterogeneity of included studies, it is not possible to determine a 'best' method for conveying probabilistic information [40]. However, whilst there might be a paucity of high quality evidence to support an unequivocal 'best' method there have been recommendations for guiding principles developed by several groups. The first, developed using an international consensus process involving researchers and patients, provided key considerations for presenting probabilities of outcomes [41]. These include:

- Use event rates to specify the population and time period

- Compare outcome probabilities using the same denominator, time period, and scale;

- Describe uncertainty around probabilities;

- Use visual diagrams;

- Use multiple methods to view probabilities (words, numbers, diagrams);

- Allow the patient to select a way of viewing the probabilities (words, numbers, diagrams);

- Allow patient to view probabilities based on their own situation (e.g. age);

- Place probabilities in context of other events;

- Use both positive and negative frames (e.g. showing both survival and death rates).

An expert consensus group further developed these IPDAS items to develop a set of guiding principles and key messages which cover eleven components of risk communication and consider what information to present and how it should be presented within tools such as patient decision aids [42]. The guiding principles range from how best to present the chance an event will occur, to use of interactive web-based platforms for delivery, and narrative methods for communication [42]. A recent study published 'good practice statements' for the development of evidence-based information communicating the effects of healthcare interventions [43]. Many of these statements would be relevant for developing information related to risk communication to support decision about trial participation. For example: using numerical formats that are easy to understand; present both numbers and words; and report absolute effects [43]. In summary, whilst there may be a paucity of high quality evidence to underpin decisions

about effective risk communication in clinical trial contexts, many of the good practice recommendations developed through empirical research provide sensible frameworks to promote informed choices, enable good quality decision making, and are unlikely to cause significant harm. As such, these guiding principles could also serve as a foundation on which to develop (and test) effective methods of risk communication within the context of clinical trials.

## Strengths and limitations

The low number of studies included for review means it is difficult to confidently make far reaching recommendations based on the findings, and the heterogenous nature of the studies mean a meta-analysis was not feasible. The studies in our review included decisions about trial participation that were hypothetical which may limit the extent to which these findings are applicable to a real world setting. Understanding and assessing risk and risk communication is pertinent to the trial phase, as the magnitude of risk is much greater in earlier phases of clinical trials; however, only one of the studies stated the trial phase being investigated. This review is, however, the first to systematically investigate risk communication within a clinical trial context. With ever increasing numbers of trials, the importance of informed consent, and yet no consistent, evidence-based format for presenting probabilistic information in a clinical trial setting, this study supports the argument for effective future research within this area.

## Conclusions

The evidence base surrounding risk communication in clinical trials indicates that there is as yet no clear optimal method for improving participant understanding, nor a clear consensus on how understanding affects willingness to participate, indicting a necessity for robust, high quality research in this area. Further research into risk communication during the informed consent process for trials, based on examples of best practice in other settings such as treatment and screening decision making, is needed to help illuminate the mechanisms underlying risk perception and understanding and provide appropriate ways to present and communicate risk in a trial context so as to further promote informed choices about participation.

## Supporting information

**S1 Checklist. PRISMA checklist.**
(DOC)

**S1 Appendix. Search strategy MEDLINE.**
(DOCX)

## Acknowledgments

The authors would like to acknowledge Cynthia Fraser for help designing and running the search strategies and Paul Manson for updating the search.

## Author Contributions

**Conceptualization:** Katie Gillies.

**Data curation:** Maeve Coyle.

**Formal analysis:** Maeve Coyle.

**Investigation:** Maeve Coyle, Katie Gillies.

**Methodology:** Katie Gillies.

**Project administration:** Maeve Coyle.

**Supervision:** Katie Gillies.

**Validation:** Katie Gillies.

**Visualization:** Katie Gillies.

**Writing – original draft:** Maeve Coyle.

**Writing – review & editing:** Katie Gillies.

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
