## [Decision Letter · Decision Letter 0]

30 Oct 2019

PONE-D-19-19791

A systematic review of risk communication in clinical trials: how does it influence decisions to participate and what are the best methods to improve understanding in a trial context?

PLOS ONE

Dear Dr. Gillies,

Thank you for submitting your manuscript to PLOS ONE. After careful consideration, we feel that it has merit but does not fully meet PLOS ONE’s publication criteria as it currently stands. Therefore, we invite you to submit a revised version of the manuscript that addresses the points raised during the review process.

The first reviewer argues that this manuscript is not suitable for publication since it cannot be regarded as a systematic review and PLOS ONE only accepts systematic reviews. At the same time, systematic reviews of ethical issues may differ from conventional medical systematic reviews. There are several methodogical ethical papers on the conduct of systematic reviews in ethics. Therefore, please revise the manuscript along these lines and clarify how the manuscript can be regarded as a systematic review.

We would appreciate receiving your revised manuscript by Dec 14 2019 11:59PM. To enhance the reproducibility of your results, we recommend that if applicable you deposit your laboratory protocols in protocols.io, where a protocol can be assigned its own identifier (DOI) such that it can be cited independently in the future. For instructions see: http://journals.plos.org/plosone/s/submission-guidelines#loc-laboratory-protocols

We look forward to receiving your revised manuscript.

Kind regards,

Rieke van der Graaf

Academic Editor

PLOS ONE

Journal Requirements:

2. Please include your tables as part of your main manuscript and remove the individual files. Please note that supplementary tables (should remain/ be uploaded) as separate "supporting information" files

4. Please amend the file type of your PRISMA checklist to a 'Supporting Information' file type.

Reviewers' comments:

Reviewer's Responses to Questions

**Comments to the Author**

1. Is the manuscript technically sound, and do the data support the conclusions?

Reviewer #1: Partly

Reviewer #2: Yes

2. Has the statistical analysis been performed appropriately and rigorously? 

Reviewer #1: N/A

Reviewer #2: N/A

3. Have the authors made all data underlying the findings in their manuscript fully available?

Reviewer #1: Yes

Reviewer #2: Yes

4. Is the manuscript presented in an intelligible fashion and written in standard English?

Reviewer #1: Yes

Reviewer #2: Yes

5. Review Comments to the Author

Reviewer #1: The authors conduct a systematic review of studies examining different methods of presenting risk and benefit information in the context of clinical trial consents. While the initial search generated a large number of prospective papers, the inclusion criteria, which specifically required the studies to be examining communication in a clinical trial context, resulted in only 7 widely varying papers (all of which relate to hypothetical clinical trials decisions) being actually examined. The authors summarize the findings of these 7 papers and provide some additional context in the discussion.

While the basic structure and writing of the paper were acceptable, I have a number of conceptual concerns.

First, this paper is really more of a review of a systematic review. Yes, there was a systematic process for identifying papers and abstracting information. But, the papers are themselves so varied that no conclusions can be drawn except at the individual study level. In other words, no new insights are gained by the set of information beyond the understanding of best practices that existed previously. Given that this journal explicitly excludes review papers, I am thus skeptical that this paper is appropriate for publication here.

The main problem is in the definition of the problem to be studied: The authors restrict themselves to papers about clinical trial contexts. The logic for doing so is based on the idea that risk communication in a clinical trial context is somehow fundamentally different than other contexts. It therefore excludes many other, very similarly structured studies examining risk communication in other kinds of decisions. While systematic reviews are very valuable way of considering a range of papers that have studied similar questions, the current set of papers are not really "similar" in important ways. Thus, the criteria is both narrow (in focusing on clinical trials situations only) and very broad (in allowing in all kinds of risk communication of very different types).

I can't really complain about the analysis because the analysis is entirely individual paper focused. Each study is summarized, but there is no real integrative analysis. Because of its narrowness, little of the risk communication lessons derived from other work are discussed. This leaves readers with the mistaken impression that little is known about how to effectively communicate the types of risk information relevant to clinical trials. What is lacking (as this paper shows) is empirical studies showing that these lessons carry over to this context. The unfortunate reality is that the lack of relevant studies is because there is little incentive for researchers to do such conceptual replication studies when the answers seem so obvious.

Given the limitations of the data, i am also concerned by the length of the discussion section and the degree that it tries to draw conclusions from single studies (which is antithetical to the concept of a systematic review).

Reviewer #2: The paper reports a systematic review of studies addressing the effects of various methods to communicate the patients about the risks of participation in RCTs of treatments. They found 7 studies after considering almost 5000 titles. Of the 7, five were RCTs (of communicating about RCTs) and two had a different design (prospective, non randomized). Additionally, NONE of these studies were done with patients who actually considered participation in a clinical trial for their own disease. Rather they asked the participants to hypothetically pretend they had a disease and decide whether they would participate in a trial of a fictional treatment for the disease. This is a limitation of the field, not of the review.

Because the studies did not use the same interventions nor measure the same outcomes, it was not possible to summarize the effects of the communications quantitatively (meta analysis), so they did a narrative summary. What this means in practice is that they summarized in extensive detail the methods and findings of each study, in alphabetical order of authors, grouped by whether the study was an RCT or not.

My perception of the paper has two high level comments.

Comment 1. The descriptions of the studies have too many details, described “telegraphically”. That is, you almost have to go read the paper to understand what was done. An example is this extract from the description of the second study described:

“three different risk/benefit formats: text (e.g. risks and benefits described as

219 number out of 100 experiencing outcome); , tables (e.g. same as text based but presented in tabular

220 format); or pictographs (e.g. a matrix of 100 rectangular blocks representing 100 children then

221 coloured to represent presence (blue) or absence (grey) of outcome). The presence or absence of a

222 risk severity graphic was also investigated. This graphic was described as having increasing risk

223 represented by blocks of different sizes and colours (yellow through red), and an arrow depicting

224 where the risk or side effect fell on the scale accompanied by a description of risk as ‘mild’, ‘moderate,

225 or ‘severe’. Risks and benefits of two drugs for post-operative pain in children were communicated in

226 one of three formats and presented as either absolute risk or incremental risk to participants.”

While I am sure these statements accurately describe the elements of the study, it is difficult for the reader of this summary to get a sense of the design – was it 3 by 2 by 3 by 2, i.e., 36 different cells? Or 12?

Comment 2. The introduction presents “participation in an RCT” as just another area in which physicians might need to communicate about risks to patients, as if it were a routine drug or an operation. The reader might ask, what is new here, it is just a different domain in which risks must be communicated. Surely we’d expect that communication aids or approaches that work for other treatment decisions would similarly work here.

But participation in a randomized clinical trial actually has a degree of complexity, uncertainty, risk much larger than choosing a treatment. While the tradeoffs between alternative treatments involve risks and value tradeoffs, the tradeoffs between a study or not doing a study involves a) a loss of control, a LACK of being informed, and b) greater uncertainty about the benefits. I say greater uncertainty, because while treatment A has its own uncertainties (will it work? Will there be side effects?), a trial in which treatment A is one of the options to which one will be assigned has all those uncertainties, plus the uncertainties about the alternative (experimental) treatment whose uncertainties are even greater, plus the uncertainty about which of the two (or more) arms one will be assigned to. Quite possibly the communication aids or approaches that work for a decision among accepted treatments with (somewhat) known success probabilities would NOT work for this more complex situation. It would be helpful for motivating the reader to engage with the paper if this difference between communicating the risks of RCTs, versus simple (?) treatments, were stated explicitly. It makes the paper more interesting.

It would also be interesting if, in describing the studies, the review noted whether and how the papers acknowledged and addressed the added complexity of the participation decision. Did they communicate ONLY about the probabilities of treatment assignment? Or did they name the good and bad outcomes that could occur with either treatment? Did they explicitly state that the uncertainty about the benefits and side effects of the experimental treatment is greater than the uncertainty about the benefits and side effects of the currently accepted, comparison treatment? That is, what in particular was the “risk communication” about?

The paper is somewhat verbose; partly in the attempt to be very clear about what is being said, but it could be tightened.

Particular suggestions. (Some of these suggestions may be too minor, or wrong. But the reviewer instructions say that there is not going to be any copy editing on the part of the journal. Your $1500 is not sufficient to buy that service.)

Abstract, line 19. “context” should be “contexts”.

Line 27. When first encountered, it is difficult to understand what “studies set within hypothetical host trials” refers to. I have interpreted it as “studies in which risk information is communicated to support the participant’s hypothetical decision to participate in a fictional trial of a fictional treatment for a disease that the participant does not have”, but that is not what the words “set within a hypothetical host trial” means.

Line 31. “In term so” should be “in terms of”.

P 4, line 50. Expand “participants approached to take part and those consented to trials” to ““participants approached to take part and those who consented to participate in trials”

Line 51, participants.

Line 57, participant should be plural.

Line 66, 67, change “risk communication with regard to public health messages, health and treatment decisions for patients, and in screening contexts” to “risk communication with regard to public health messages, health [behavior?], and treatment and screening decisions for patients”. I guess “screening” may be done on people who are not technically “patients”, but so often it is done in primary care clinics.

Line 88, change “findings from work from our group has shown” to “findings from our group have shown”?

Line 91, change “highlighted patients preferences” to “highlighted that patients’ preferences”.

Lines 93 and 94, move “from treatment and screening decisions” to after “decision making” in the next line, and change “from” to “about”.

Last sentence before “Methods”. Participants should be possessive.

Methods. I am not sure what an “explanatory” study is. I don’t think it is an “exploratory” study (though that is used in next paragraph, but those studies were excluded). Does it mean, “a study of ways of explaining something to a patient”? If that is what it means, I don’t think the phrase “explanatory study,” without more elaboration, will be understood.

Style. “were included” was used three times in this one paragraph.

Lines 199 and 120, I don’t think it needs both “exploratory studies” and “that have explored”. Maybe drop the first.

Lines 124 and 125. What is a “senior information officer”? a modern librarian?

Line 129. Say May 10th is in 2019.

Line 138. “and progressed for data extraction procedures”? does than mean, “were followed by,” “led to”, “progressed to”?

Line 142. Given the immediately preceeding sentence, change “The seven studies deemed eligible and identified for data extraction were summarized” to “The seven selected studies were summarized”.

146. I’m not sure the Journal’s conventions. I would add commas around “i.e.” and “e.g.”, through out, “by content of intervention, i.e., probability” not “by content of intervention i.e. probability”

Data extraction and management.

Line 151. Data were, not data was.

Lines 151 and 152. Change “summary data was extracted regarding the host trial for each study and summarised in table form” to “summary features of the host trial for each study were summarised in table form”. Of course you had to extract it before summarizing it, but it need not be said.

Line 155. Space after the comma.

4 instances of i.e. without surrounding commas in this one paragraph.

Line 160. Change “satisfaction with communication method; and level of decisional conflict” to “satisfaction with communication method, and level of decisional conflict”

Lines 168 169. Here might be a place to give an example of how a study of communication about the risks of RCTs might be “set within a hypothetical host RCT”.

Line 170, drop unnecessary words, in the context. Change “The seven included studies that explored interventions to explore risk communication during informed consent varied by design” to “The seven included studies had various designs”.

Line 172. Could “prospectively designed” be simply “prospective”?

P 9, line 179. “an” not “and”

Line 180, “included” is ambiguous. Does it mean one study was only of parents, or one study had not only patients deciding for themselves but it also included some parents deciding for children?

Lines 186-188. It was unclear to me at first that these were the authors of the studies you were summarizing; at first I thought they were references for the kinds of method used, e.g., probability presentation, risk framing, risk interpretation. Be more clear.

194. Hypothetical is unclear. Is it that the participant is hypothetically making a decision about a real study, or is hypothetically making a decision about participation in a hypothetical or fictional study? The word is too vague for the many possibilities.

194. But since you’ve already told us all the studies involve pretending to decide about a fictional study, if I interpret correctly, you do not need to repeat it every time you describe a study. It just confuses the reader.

Line 197. Is the reference a cite of an included study, or of the paper describing numerical values associated with words like “common” in an official EU document? Or of an earlier study of the EU prescribed communication language?

Line 199. Add “were” before “asked”.

P 10, lines 208 and 209. Put quotes around the terms common, uncommon, and rare.

Line 217. Drop “t”.

Line 219, and extra comma.

Line 225. 2 by three by 2 design? Or did each get info about both drugs in the same format?

Line 230. Move “(66.55) back to after “gist understanding”.

234. orphan “).”

239. Comma after “respectively)”?

239. Comma after graphics.

252. space after 8.3. comma after “scale”.

259 to 262. The instructions in this study were, ‘Out of 100 people whose lives would likely be cut short by heart disease and

260 begin taking this drug, we expect that 95 will show substantial improvements in their chance of

261 survival and 5 will show no improvement in survival’. Do you want to comment on how vague and uninformative that is?

268 to 271. “The majority of participants (59%) chose to take part in the trial when outcomes were framed as losses, however a greater majority of the ‘gain’ group (65%) declined to participate. When both framings were presented, 62% of participants chose to participate, making a similar choice to the ‘loss’ group.” I find the switch in reference to be confusing. Who cares the relative size of the majority? Keep it consistent: “The majority of participants (59%) chose to take part in the trial when outcomes were framed as losses, while only 35% of the ‘gain’ group chose to participate. When both framings were presented, 62% of participants chose to participate, making a similar choice to the ‘loss’ group.”

278 needs a period at the end.

313 to 317. This discusses inaccuracy of patient mapping of numbers onto EU definitions of probability words. To interpret this, one would need to know what all the options are. Are they missing by one category, or are they several off? But the larger question is, what is the relevance of this to communicating about the risks of participating in randomized clinical trials? This seems a general problem with any risk communication. Of course, it is part of the whole comprehension process.

Line 326. Change “for only benefit” to “only for benefit”?

Line 326. I’m not sure what “saw” means. Was this a study intervention, or a participant behavior, picking up on an implication that is present? I think it is the former. I think it is a situation of: show one of three formats; see how many consented. If that is what it is, then you are describing it backwards, in terms of causality. “of those who consented, X saw a format” is the reverse order. The interesting question is, “of those who were shown x format, who saw x format, what proportion consented? I think this needs to be restated.

Line 327. Not clear what it was “to see neither”

Line 339. “aspect” should be plural.

345. “identified” seems an unnecessary word.

354. “deemed”. Is this that the participants reported the pictographs as more helpful, or that the authors of the reviewed studies concluded based on their data that the pictographs were more helpful. Who is “deeming”.

355. Not clear what the distinction is between a pictograph and a severity graphic, why is the one good and the other bad?

In this discussion, I feel a need for the authors to more authoritatively criticize the reviewed studies, not just to summarize them. I wonder if it would be helpful to have a model of comprehension of risk communication, e.g., the required steps: a) comprehend the scale, b) use the info to interpret one’ situation and guide action. Of course, if they fail the first step the second step has little chance of happening – unless they get it from alternative clues like verbal summary of gist. Perhaps you get at that in line 371.

Line 363. Change “and verbal conditions, however it did find” to “and verbal conditions; however, it did find”

Line 371. Change “as well as comprehending the concept” to “nor to comprehend the concept”. Preserves parallel with the previous clause.

373 to 374. It would help if the manuscript more clearly separated the paragraphs. This is an example where it is hard to tell.

Line 374. Put “The” before “Sutherland”.

Line 379. “generally”? should that be “usually”? Mathematically, “generally” means always true.

382. I think it is “their best interest” rather than plural, but I’ve seen it both ways.

390. “when gains were framed as gains”? “when outcomes were framed as gains”.

403. sheets = leaflets. If you don’t say that, the reader does not know what “PIL” means.

Lines 406 407. This list is hard to interpret, the elements do not seem parallel, and it is hard to tell if “understanding of” applies only to “trial processes”, or to all three elements in the list.

416. Not sure the meaning of “its evidenced importance”. It’s evident importance (on the face of it), or its proven importance (proven by lots of previous studies, including the evidence reported by study 17)?

Line 426. I’m not sure if agreement should be “number… means” or “number of studies… mean”.

428.”hypothetical nature of the studies”. But these were real studies, actually done. Need a different way to talk about this. And it is ambiguous if you mean “the 7 studies we reviewed” or “the stimulus studies, presented to the participants in the studies”.

Line 432. I don’t think “however” obeys the same rules as “but”. So I think the punctuation should be not “earlier phases of clinical trials, however only one of the studies” but “earlier phases of clinical trials; however, only one of the studies”.

Line 433. I think however needs to be set off on both sides by commas.

Line 440. What does “it” refer to?

443. “most useful” compared to what, or out of what set of options for this particular situation? Or do you mean more useful in this situation than in other situations where it might be used?

444. I think it is “complementary” not “complimentary”.

In the reference list, words are capitalized inconsistently. Some titles have all the words capitalized (appropriate for book titles but not paper titles) while others have just the first word capitalized. Be consistent, and consistent with the journal standards.

Reference 25 needs a period between the title and the journal name.

Reference 38. Laura siminoff’s paper had only one page?

Figure 1. I note that you found 3 systematic reviews – how did they differ from the present systematic review? Should they be included in your literature review (not just in your characterization of your systematic review’s rejects).

Table 2. Why a date for Tait but not the others.

6. PLOS authors have the option to publish the peer review history of their article (what does this mean?). If published, this will include your full peer review and any attached files.

Reviewer #1: No

Reviewer #2: Yes: Robert M. Hamm

---

## [Author Response · Author response to Decision Letter 0]

19 Dec 2019

Please see uploaded 'cover letter' which contains a detailed response to reviewers comments.

---

## [Decision Letter · Decision Letter 1]

4 Jul 2020

PONE-D-19-19791R1

A systematic review of risk communication in clinical trials: how does it influence decisions to participate and what are the best methods to improve understanding in a trial context?

PLOS ONE

Dear Dr. Gillies,

Thank you for submitting your manuscript to PLOS ONE. After careful consideration, we feel that it has merit but does not fully meet PLOS ONE’s publication criteria as it currently stands. Therefore, we invite you to submit a revised version of the manuscript that addresses the points raised during the review process.

Generally the reviewers are happy with the review. Much of the problem focuses on the nature of the review. PLOS ONE accepts systematic reviews but not narrative reviews. However, reviewer #1 raises an important point that the pure systematic review that concludes that there isn't enough evidence is useful but not very helpful. On the other hand a more narrative review might be more helpful. I think that there may be a compromise here. Maintain the systematic review but have a more detailed analysis of the topic in the discussion. Thus, it is a systematic review with a more comprehensive analysis.

We look forward to receiving your revised manuscript.

Kind regards,

Dermot Cox

Academic Editor

PLOS ONE

Reviewers' comments:

Reviewer's Responses to Questions

**Comments to the Author**

1. If the authors have adequately addressed your comments raised in a previous round of review and you feel that this manuscript is now acceptable for publication, you may indicate that here to bypass the “Comments to the Author” section, enter your conflict of interest statement in the “Confidential to Editor” section, and submit your "Accept" recommendation.

Reviewer #1: (No Response)

Reviewer #2: (No Response)

2. Is the manuscript technically sound, and do the data support the conclusions?

Reviewer #1: Partly

Reviewer #2: Yes

3. Has the statistical analysis been performed appropriately and rigorously? 

Reviewer #1: N/A

Reviewer #2: I Don't Know

4. Have the authors made all data underlying the findings in their manuscript fully available?

Reviewer #1: Yes

Reviewer #2: Yes

5. Is the manuscript presented in an intelligible fashion and written in standard English?

Reviewer #1: Yes

Reviewer #2: Yes

6. Review Comments to the Author

Reviewer #1: This revised manuscript reports a systematic review of studies examining different methods of presenting

risk and benefit information in the context of clinical trial consents. The authors required the studies to be examining communication in a clinical trial context, resulting in only 7 widely varying papers which are then summarized one at a time. The primary conclusion is that there is little extant guidance for presenting risk and benefit information to guide clinical trial participation decisions.

In my previous review, I expressed significant concerns about the authors' framing of the scope of this review. In particular, the restriction to studies whose scenarios / tasks could only be about clinical trial decisions results in a small, heterogeneous set of papers and does not consider the relevance of structurally similar papers addressing other contexts such as treatment or screening decisions. The resulting manuscript thus is more descriptive than integrative, and it conveys a primary message of "we don't know", which is literally true but practically false (we do know a lot that has been demonstrated to generalize across decision contexts).

In their response to my review, the authors reiterated that their work aligns to the process of a systematic review. Such was not actually my concern, and I agree that their work does in fact follow that process. My concern is that, in this context, a true systematic review is less valuable to the field and less useful to practitioners than a review that is more integrative.

The issue is actually simple: I fully agree that actual clinical trial participation decisions are different in some important ways from actual treatment decisions. I am quite skeptical that _hypothetical_ clinical trial decisions (of the type studied in the reviewed papers here) are actually that different from _hypothetical_ treatment decisions because many of the differences in real-world mindsets are gone, replaced by the common experience of reading hypothetical scenarios. More importantly, I fundamentally disagree with the idea that the process of _communicating_ _risks_ is so categorically different in clinical trial contexts that the exclusion of all other studies, and the resulting conclusion that little is known about how do risk communication well in this context, is justified. The authors do not provide a clear argument how the comprehension task, the interpretation task, and/or the integration tasks associated with receiving and using risk information in this context are different from those same tasks in, say, a decision support tool for treatment or screening decision making.

The authors have made several relatively minor revisions that acknowledge that "lessons from effective risk communication in a treatment and/or screening context can provide examples of best practice that could be used for those developing PILs for patients considering clinical trial participation" and that future research could use "the best practice examples from treatment decision making as a starter" They also now refer to one of a number of relevant risk communication reviews to begin orienting readers to that literature. I thank the authors for these steps, as they do mitigate my concerns somewhat. But, I am not sure that they are enough to offset the primary conclusions (as still stated in the abstract) that "there is as yet no clear optimal method for improving participant understanding," despite lots of evidence from other non-included studies that certain approaches are likely to be more effective than others.

My preference would be for an option that I suspect the authors (and perhaps the journal) would reject: To create a kind of hybrid paper, one that does the systematic identification of papers and notes the dearth of relevant empirical evidence (which, to me, is the primary value of this paper) but then complements it with a much more detailed and somewhat speculative discussion of _specific_ principles of good risk communication evident in the broader literature that are relevant in clinical trial contexts. This type of paper could be quite valuable to many audiences in ways that I fear that the current manuscript will not be.

In net, I don't have much in terms of small comments. The authors have been generally responsive to the detailed comments provided by other reviewers, and they have taken some steps towards addressing my own comments. I just remain quite concerned that the takeaway message of this paper is not particularly helpful (and may be harmful) in guiding improvements in risk communications in clinical trial documents.

Reviewer #2: I read the review with better comprehension this time, thanks for working to organize and clarify it.

There are some verbal expressions that confuse me. It may be that the UK and the US are “two countries divided by a common language”. One is the characterization of the studies as having been “nested in” or “hosted in” or even “within” a hypothetical trial. From reading the paper the situation is just that the studies were done with a cover story or situation (a trial for a disease that the participant does not actually have) and the information about the situation is varied. The “hosting” and “nesting” and “within” convey that there is something bigger going on, say a study with multiple interviews over time, and that an incidental add on is that on one occasion among the 4 questionnaires filled out that day is the booklet that contains the study of risk communication. But I don’t find any evidence for this.

Is this part of the way people in the UK talk is it an idiosyncratic choice of words just this once? If the latter, it is confusing for this reader and it seems to add unnecessary work to the task of reading the review. There was some complaint about this last round, and the promise in the author’s response that it had been fixed, but it does not seem to have been addressed.

On the presentation of the reviewed studies’ findings, in the Results section. I note that when comparisons are made, the statistic presented is the p value. Only a few times was there an effect presented (X% versis Y%). Quite possibly those numbers were present in the previous version and have been removed because of the complaint of “too much unreadable information”. So I need to be careful here.

I know that the social science statisticians, such as the American Psychological Society (to which the authors need not have any loyalty), ask for an effect, a degree of freedom or N, as well as the p value. And then there are the new standards of presenting the effect estimate and confidence intervals. But these are for primary studies. For reviews, however, the same general ideas are still relevant for the reader to understand how much weight to put on a study.

Degrees of freedom. While the text need not have a df next to every p, how about when the study is first described, saying how many people were in it, in each group? Then we can know if it was 10 or 1000, and form opinions accordingly.

Effect size. Of course there is the issue of statistical significance compared with clinical or practical significance. If we knew the N was huge and the effect was p = .05, we might suspect the effect is actually small and of little practical importance (in the absence of the text giving the effect size). But really, for at least half of the “(p < .xxx)” statements, providing a simple idea of what the compared means were might be a great help. If probabilities, we know what those scales mean. Sometimes if it really would help understand the effect, you might need to explain the scale.

Particular suggestions.

Abstract. “participants” needs to be plural possessive in this: “studies that aimed to support potential trial participants decision”.

P 4. First para, I suggest changing “should include explanations of the ‘reasonably foreseeable risks or inconveniences’ and expected benefits, and where there are no clinical benefits to the participants, they must be made aware of this” to “should include explanations of the ‘reasonably foreseeable risks or inconveniences’, expected benefits, and where there are no clinical benefits to the participants, they must be made aware of this”

P 5, line 78. “Decision” should be plural.

I suggest changing “Conditional altruism is the concept that participation in the trial will benefit society but there is a need for a benefit (which is influenced by perception of risk) for self to be tied in” to “Conditional altruism is the concept that participation in the trial will benefit society but there must be a benefit (which is influenced by perception of risk) for self”.

Suggestion: change clumsy punctuation from “trade-offs between risk and benefit in a trial involve; layers of complexity in addition to those for treatment such as: loss of control over which treatment they receive; and potentially greater uncertainties as often participants have to consider the risks and benefits of a minimum of two competing treatments” to “trade-offs between risk and benefit in a trial involve layers of complexity in addition to those for treatment, such as loss of control over which treatment they receive and potentially greater uncertainties, as often participants have to consider the risks and benefits of a minimum of two competing treatments”

P 6, line 101. Add “that” after “highlighted”.

Line 121 “within information presented” These words seem redundant. Can they simply be deleted?

P7, line 126. Add “studies of” after “include”.

Lines 130-131. What does this mean? I don’t understand what kinds of studies were excluded. “More eStudies investigating participants perceptions of receiving risk communication…”

Is an “eStudy” a thing? Why exclude it?

Also, “participants” needs to be plural possessive.

I understand that you consider a “Senior Information Officer” to be distinct from a “librarian”, as stated in response letter. But simply leaving it as such does not solve the problem that for people in different university systems, there is no such thing. Could you explain it in a phrase? Is it like a police officer, the “fact police”. That would be interesting. In my area, “public information officers” are public relations flacks, and then there are “information systems security officers”, but I would not go to either one for help with a library search.

P 9. Line 183. “…the studies asked participants to imagine they were being recruited into a host clinical trial, provided brief information about the host trial (such as clinical population, intervention, comparator, outcomes, etc), then provided various formats of risk communication (such as verbal or numerical descriptors) and assessed relevant outcomes.” “assessed relevant outcomes” is ambiguous. Is this part of the participant’s task (to assess relevant outcomes that are described), or is “assessed” an adjective describing a part of the materials that the participants are shown; where someone else has assessed the outcomes (probably verbally, versus numerically) and the participant looks at it and does some other judgment (e.g, whether to sign up for the trial). I think it is the latter. If so, changing “assessed” to “assessments of” would make it clearer.

Line 186. An example of my problem with “nested in”. How about “in the context of” or even just “about”?

In 190, again, “set within a trial” just means that the context of the vignette was a non drug intervention rather than a drug.

P 10. Line 198, explanation of structure of upcoming review. Again, “embedded” is a confusing word that could simply be dropped!

I did not know whether the two groupings --first design, then content of intervention -- were nested (in the statistical sense) or crossed. It turns out that the “review” section groups them by design with intervention nested within; And then the discussion covers them again, but this time by intervention (all designs). The material is readable and accessible, when actually reading the two sections. This “heads up” could be less confusing if it dispensed with the “first, then” and just said the “The final seven studies were grouped according to the study design (RCT or prospective cohort) and the topic of the described intervention: ‘probability presentation’ (22, 23 ), ‘risk framing’ (24, 25, 26, 27), and ‘risk interpretation’ (28).”

Lines 216 to 219. There are three instances where an extra “9” has been inserted.

Line 216. The list might be changed from “…also perceived the risk to health to be higher (9p<0.0001), benefit to be lower (9 p=0.03), and were…” to ““…also perceived the risk to health to be higher (9p<0.0001) and the benefit to be lower (9 p=0.03), and were…”

Line 232. There is an extra “,” in the “p <” parenthesis. I think this indicates that you went through and removed them all. Sorry for my contradictory advice.

P 12, line 260. Could you add a comma after “pain” in “trial testing two drugs for post-operative pain one a standard treatment and the other”?

Line 261. There is an orphan “t”.

Line 265. Comma after “risk reduction”

Line 266. Change “parent show” to “parents who”. Autocorrect sometimes errs.

Line 271. Add “to be” before “more likely to agree”.

The Tait study seemed to manipulate whether disadvantageous information was presented or hidden. Some of the other studies manipulate how disadvantageous (and advantageous) info is described, framed, but they present essentially the same in every condition. Although you touch on this briefly in the discussion (pp 17-18), it seems to be a very important distinction between the kinds of manipulation in these studies. The implications of the findings are different. When you find the (obvious) that hiding the disadvantages makes people more likely to choose an option, that informs the researcher/sponsor’s decision whether to lie and obfuscate in a propagandistic marketing manner. When you find that the same information, framed differently, influences option choice, that informs the researcher/sponsor’s decision whether to “nudge” in a libertarian paternalistic manner https://en.wikipedia.org/wiki/Libertarian_paternalism.

P 14, line 299. “Participants were presented with information in one of six combinations in a pre-specified order.” First, you might put “the” before “six,” since you have already described the 2 (order) by 3 (content) design. But I thought that each person got only one condition? What is the function of “in a prespecified order” in this sentence? If it is about the two sequences, you have already incorporated it. Is there a different order? Of course there is a “procedure” that has a lot of steps in order: meet, qualify, consent, instruct, and so on. Usually that need not be said. Could you drop “in a pre-specified order”?

Line 308. You could drop the conversational “When it came to presentation preferences” and the reader would still know exactly what you are talking about.

Line 318 add “and” between “risk” and “benefit”.

Line 319. Add “noted only” after 27%.

P 18, line 408, change punctuation and one word to “An analysis of PILs used in clinical trials by Gillies et al (2011) found that explaining trial processes, presenting probabilities, and expressing values were consistently poor across all PILs when assessed using an informed consent evaluation instrument”.

P 19. Lines 448-449, change “indicates that there is as yet not clear optimal method for improving participant understanding, or clear consensus on how” to “indicates that there is as yet not a clear optimal method for improving participant understanding, nor a clear consensus on how”

Line 451, change “within trials” to “about trials” or “during trials”? “within” evokes my ANOVA framework.

7. PLOS authors have the option to publish the peer review history of their article (what does this mean?). If published, this will include your full peer review and any attached files.

Reviewer #1: No

Reviewer #2: **Yes: **Robert M. Hamm

---

## [Author Response · Author response to Decision Letter 1]

14 Sep 2020

Please see uploaded response letter which contains a detailed response to reviewers comments.

---

## [Decision Letter · Decision Letter 2]

14 Oct 2020

PONE-D-19-19791R2

A systematic review of risk communication in clinical trials: how does it influence decisions to participate and what are the best methods to improve understanding in a trial context?

PLOS ONE

Dear Dr. Gillies,

Thank you for submitting your manuscript to PLOS ONE. After careful consideration, we feel that it has merit but does not fully meet PLOS ONE’s publication criteria as it currently stands. Therefore, we invite you to submit a revised version of the manuscript that addresses the points raised during the review process.

The reviewers feel that the paper is much improved but they still have a few minor issues that need to be addressed.

We look forward to receiving your revised manuscript.

Kind regards,

Dermot Cox

Academic Editor

PLOS ONE

Reviewers' comments:

Reviewer's Responses to Questions

**Comments to the Author**

1. If the authors have adequately addressed your comments raised in a previous round of review and you feel that this manuscript is now acceptable for publication, you may indicate that here to bypass the “Comments to the Author” section, enter your conflict of interest statement in the “Confidential to Editor” section, and submit your "Accept" recommendation.

Reviewer #1: (No Response)

Reviewer #2: (No Response)

2. Is the manuscript technically sound, and do the data support the conclusions?

Reviewer #1: Partly

Reviewer #2: Yes

3. Has the statistical analysis been performed appropriately and rigorously? 

Reviewer #1: N/A

Reviewer #2: Yes

4. Have the authors made all data underlying the findings in their manuscript fully available?

Reviewer #1: Yes

Reviewer #2: Yes

5. Is the manuscript presented in an intelligible fashion and written in standard English?

Reviewer #1: Yes

Reviewer #2: Yes

6. Review Comments to the Author

Reviewer #1: I thank the authors for adding the section towards the end of the paper alluding to the fact that other risk communication guidance does exist that plausibly informs what can be done in the clinical trial contexts.

One concern: The authors need to be careful in their language to draw distinctions between what is known about risk communication in clinical trials (this work) vs. risk communication in general, and within the latter, between risk communication intended to motivate vs. risk communication intended to inform. This latter distinction is often conflated in the literature, with problematic results.

For example, the authors cite a NICE report (reference 40). But their presentation notes that the outcome used to guide this review was "effective strategies for improving health behavior outcomes" -- in other words persuasion. Such is confirmed later on when it is noted that a strategy "was more persuasive for adoption of certain behaviors." This criteria is precisely NOT what one wants to use to evaluate strategies for use in clinical trial contexts where persuasion would be ethically inappropriate. Thus, I recommend removal of this paper and its recommendations from this analysis. References 41/42/43, by contrast, focus on more analogous medical risk communication contexts and are more appropriately included.

On a related note, later in that section, the authors summarize stating "In summary, whilst there may be a paucity of high quality evidence to underpin decisions about effective risk communication many of the good practice recommendations developed through empirical research provide sensible frameworks to promote informed choices, enable good quality decision making, and are unlikely to cause significant harm." This first phrase needs to be modified to focus on clinical trial contexts, as I would not characterize the general risk communication field as having a "paucity of high quality evidence", only evidence that aligns tightly with the clinical trial context. Thus, rewording to state "decisions about effective risk communication IN CLINICAL TRIAL CONTEXTS, many of the ..." would more accurately reflect the state of the field.

Reviewer #2: Thanks for careful edits and explanations. I have only minor wording suggestions.

Line 42, “potential for learning from the evidence on risk communication from treatment and screening,” change “from” to “in”.

Line 132 or 133, need a space between the sentence.

Line 166, “the host trial (i.e. the trial the potential participants were begin asked to consider participation in)”, that should be “being” not “begin”.

Lines 2016 to 218 “Those who received only verbal descriptors were significantly less satisfied

217 with the information than those who also had the numerical values when asked to rate on a scale from

218 1 to 6 (p=0.03).”

This is slightly ambiguous. Are the 1 to 6 numbers the “numerical values”? Of course not. But it would be clearer rearranged to “When asked to rate on a scale from 1 to 6 (p=0.03), those who received only verbal descriptors were significantly less satisfied with the information than those who also had the numerical values.”

Lines 286-287. I suggest changing “and the ‘risk’ group involved procedures

287 that were described as having a high risk of injury” to “and for the ‘risk’ group procedures

287 were described as having a high risk of injury”

Line 401. “decisions in ways that are not consistent with the individuals values and preferences”. Individuals should be possessive. I’m not sure if singular or plural.

439 and 440. Change “whilst the evidence on the other strategies was less clear it was not considered to be harmful.” To “whilst the evidence on the other strategies was less clear, they were not considered to be harmful.”

441. “non-numeric, visual, and statistical formats the evidence as inconsistent with regard to how” was, not as.

447-448. “Interestingly, this review showed that presenting absolute risk

448 reduction were better at maximising accuracy” was, not were.

475 “high quality evidence to underpin decisions about effective risk communication many of the good” put a comma after “communication”.

482-483 “The low number of studies included for review means it is difficult to confidently make far reaching

483 recommendations based on the findings” Change “as yet not” to “as yet no” or to “as yet not a”.

7. PLOS authors have the option to publish the peer review history of their article (what does this mean?). If published, this will include your full peer review and any attached files.

Reviewer #1: No

Reviewer #2: **Yes: **Robert M Hamm

---

## [Author Response · Author response to Decision Letter 2]

16 Oct 2020

Reviewer #1:

1 I thank the authors for adding the section towards the end of the paper alluding to the fact that other risk communication guidance does exist that plausibly informs what can be done in the clinical trial contexts.

One concern: The authors need to be careful in their language to draw distinctions between what is known about risk communication in clinical trials (this work) vs. risk communication in general, and within the latter, between risk communication intended to motivate vs. risk communication intended to inform. This latter distinction is often conflated in the literature, with problematic results.

For example, the authors cite a NICE report (reference 40). But their presentation notes that the outcome used to guide this review was "effective strategies for improving health behavior outcomes" -- in other words persuasion. Such is confirmed later on when it is noted that a strategy "was more persuasive for adoption of certain behaviors." This criteria is precisely NOT what one wants to use to evaluate strategies for use in clinical trial contexts where persuasion would be ethically inappropriate. Thus, I recommend removal of this paper and its recommendations from this analysis. References 41/42/43, by contrast, focus on more analogous medical risk communication contexts and are more appropriately included.

Response

The reviewer raises a very important point abut the different intended purposes of various forms of risk communication. On refection we also agree that including the NICE report we identified is misleading as it has a basic assumption that the intended purpose of the communication is to persuade or modify behaviour. As the reviewer highlights, this of course may not be appropriate when considering decision about participation in a clinical trial. Therefore we have removed this section of text. See edits page 19 lines 435-447.

2 On a related note, later in that section, the authors summarize stating "In summary, whilst there may be a paucity of high quality evidence to underpin decisions about effective risk communication many of the good practice recommendations developed through empirical research provide sensible frameworks to promote informed choices, enable good quality decision making, and are unlikely to cause significant harm." This first phrase needs to be modified to focus on clinical trial contexts, as I would not characterize the general risk communication field as having a "paucity of high quality evidence", only evidence that aligns tightly with the clinical trial context. Thus, rewording to state "decisions about effective risk communication IN CLINICAL TRIAL CONTEXTS, many of the ..." would more accurately reflect the state of the field.

Response

Edited accordingly. See page 20 line 477.

Reviewer #2

3 Line 42, “potential for learning from the evidence on risk communication from treatment and screening,” change “from” to “in”.

Response

Edited accordingly

4 Line 132 or 133, need a space between the sentence.

Response

Edited accordingly

5 Line 166, “the host trial (i.e. the trial the potential participants were begin asked to consider participation in)”, that should be “being” not “begin”.

Response

Edited accordingly

6 Lines 2016 to 218 “Those who received only verbal descriptors were significantly less satisfied

217 with the information than those who also had the numerical values when asked to rate on a scale from

218 1 to 6 (p=0.03).”

This is slightly ambiguous. Are the 1 to 6 numbers the “numerical values”? Of course not. But it would be clearer rearranged to “When asked to rate on a scale from 1 to 6 (p=0.03), those who received only verbal descriptors were significantly less satisfied with the information than those who also had the numerical values.”

Response

Edited accordingly

7 Lines 286-287. I suggest changing “and the ‘risk’ group involved procedures

287 that were described as having a high risk of injury” to “and for the ‘risk’ group procedures

287 were described as having a high risk of injury”

Response

Edited accordingly

8 Line 401. “decisions in ways that are not consistent with the individuals values and preferences”. Individuals should be possessive. I’m not sure if singular or plural.

9 439 and 440. Change “whilst the evidence on the other strategies was less clear it was not considered to be harmful.” To “whilst the evidence on the other strategies was less clear, they were not considered to be harmful.”

Response

In response to reviewer 1’s comment this section has now been deleted.

10 441. “non-numeric, visual, and statistical formats the evidence as inconsistent with regard to how” was, not as.

Response

Edited accordingly

11 447-448. “Interestingly, this review showed that presenting absolute risk

448 reduction were better at maximising accuracy” was, not were.

Response

Edited accordingly

12 475 “high quality evidence to underpin decisions about effective risk communication many of the good” put a comma after “communication”.

Response

Edited accordingly

13 482-483 “The low number of studies included for review means it is difficult to confidently make far reaching

483 recommendations based on the findings” Change “as yet not” to “as yet no” or to “as yet not a”.

Response

Edited accordingly. However, we believe the section the reviewer recommended editing was on line 496 not lines 482-483.

---

## [Editor Report · Decision Letter 3]

30 Oct 2020

A systematic review of risk communication in clinical trials: how does it influence decisions to participate and what are the best methods to improve understanding in a trial context?

PONE-D-19-19791R3

Dear Dr. Gillies,

We’re pleased to inform you that your manuscript has been judged scientifically suitable for publication and will be formally accepted for publication once it meets all outstanding technical requirements.

Kind regards,

Dermot Cox

Academic Editor

PLOS ONE
---

## [Editor Report · Acceptance letter]

5 Nov 2020

PONE-D-19-19791R3 

A systematic review of risk communication in clinical trials: how does it influence decisions to participate and what are the best methods to improve understanding in a trial context? 

Dear Dr. Gillies:

I'm pleased to inform you that your manuscript has been deemed suitable for publication in PLOS ONE. Congratulations! Your manuscript is now with our production department. 

Kind regards, 

on behalf of

Dr. Dermot Cox 

Academic Editor

PLOS ONE